# A short-term memory trace persists for days in the mouse hippocampus

Maha E. Wally [1,2,3,4], Masanori Nomoto [1,2,3], Kareem Abdou[1,2,3,5], Emi Murayama[1,2,3] &
Kaoru Inokuchi [1,2,3✉]

Active recall of short-term memory (STM) is known to last for a few hours, but whether STM has long-term functions is unknown. Here we show that STM can be optogenetically retrieved at a time point during which natural recall is not possible, uncovering the long-term existence of an STM engram. Moreover, re-training within 3 days led to natural long-term recall, indicating facilitated consolidation. Inhibiting offline CA1 activity during non-rapid eye movement (NREM) sleep, N-methyl-D-aspartate receptor (NMDAR) activity, or protein synthesis after first exposure to the STM-forming event impaired the future re-exposure-facilitated consolidation, which highlights a role of protein synthesis, NMDAR and NREM sleep in the long-term storage of an STM trace. These results provide evidence that STM is not completely lost within hours and demonstrates a possible two-step STM consolidation, first long-term storage as a behaviorally inactive engram, then transformation into an active state by recurrence within 3 days.

---

[1] Research Center for Idling Brain Science, University of Toyama, Toyama 930-0194, Japan. [2] Department of Biochemistry, Graduate School of Medicine and Pharmaceutical Sciences, University of Toyama, Toyama 930-0194, Japan. [3] CREST, JST, University of Toyama, Toyama 930-0194, Japan. [4] Pharmacology Department, Faculty of Pharmacy, The British University in Egypt, Cairo 11837, Egypt. [5] Department of Biochemistry, Faculty of Pharmacy, Cairo University, Cairo 11562, Egypt. ✉email: inokuchi@med.u-toyama.ac.jp

Short-term memory (STM) is formed by the delivery of a weak stimulus while long-term memory (LTM) is formed by a stronger and more enduring stimuli[1]. Moreover, STM is known to last for only a few hours, whereas LTM is subjected to a consolidation process that allows it to remain for long periods of time[2–5]. However, it is not yet clear whether STM has long-term effects beyond this time.

Engram cells are neuron populations in which memories are stored, and their specific reactivation leads to individual memory retrieval[6,7]. Recent advances and new genetic technologies enabled the use of immediate early genes[8], such as c-fos, as markers of neuronal activity to visualize and specifically label the cellular ensembles constituting the engram[9]. This gave rise to a wide range of studies that manipulated memory engrams through erasure[10], stimulation[11], or even creation of false memories[12,13]. Engram cells were also shown to be involved in memory allocation and memory association[14–16]. Furthermore, engram studies enabled the identification of the synaptic correlate of specific memories[17,18]. Two major types of engrams have been defined: the active engram, which can be naturally recalled, and the silent engram, which cannot be retrieved by natural cues, but can be activated by artificial stimulation[19,20]. Silent engrams have been shown to be present in cases of retrograde amnesia[21,22] and early-onset Alzheimer's disease[23]. Moreover, silent engrams are involved in the reorganization of remote memories between the hippocampus and cortex through system consolidation[19,24]. However, all engram-based studies have focused on several types of LTM, and none have investigated the existence of such populations in case of STM.

Memory consolidation process depends on several factors such as new protein synthesis, whereby the inhibition of post-learning protein synthesis blocks LTM but not STM retrieval[25,26]. It also depends on N-methyl-D-aspartate receptor (NMDAR) activation, where synaptic plasticity has been shown to be integral for memory consolidation[5,27,28]. Finally, consolidation has also been shown to occur after a period of post-learning sleep[29–34] which consists of two main stages, non-rapid eye movement sleep (NREM) characterized by slow delta waves, spindles and sharp wave-ripples (SWRs), and rapid-eye sleep (REM) characterized by fast theta rhythms[30]. It has been shown that hippocampal SWRs, which occur during NREM sleep, and theta rhythms, which occur during REM sleep, are both critical for the consolidation of spatial memories[35,36].

We hypothesized that, beyond the few hours of active STM recall, a trace of STM may continue to exist, but is unable to be naturally retrieved. However, its activation could have long-term effects, nevertheless. We investigated this hypothesis using optogenetic recall of an STM-forming event, which unveiled the presence of an STM engram stored within the hippocampal circuit. Using a simple behavioral approach, we demonstrate that the STM trace provides a template that facilitates consolidation upon future re-exposure to the same event within a specific period of time. Using pharmacological manipulations, we investigated the mechanisms behind the long-term storage of this STM trace. Contrary to the previously accepted concept, STM triggers new protein synthesis for the long-term storage of its trace. Furthermore, we show that NMDAR is involved in facilitating STM consolidation. Finally, we highlight a necessity of post-learning sleep in preserving the long-term availability of the STM trace in a similar fashion to its consolidation of LTM trace using specific optogenetic silencing of offline activity during post-learning NREM sleep. Taken together, these results modify our current understanding of the STM and provide mechanistic insights into its potential long-term storage and effects.

## Results

### LTM and STM in novel object location (NOL) task

We used the NOL behavioral paradigm[23], a hippocampal-dependent spatial memory[37], as a learning task. Mice freely explored the location of two objects during a training session, and then during the test session one object was moved to a new location while the other remained in its original location (Supplementary Fig. 1). When mice recall the memory of object location, their exploration of the moved object is greater than that of the unmoved object. However, an equal exploration of the two objects indicates that the mice have not discriminated between the new and old locations and have been unable to retrieve the memory. A previously established approach[38] of using two training protocols for the generation of LTM and STM was conducted as follows: a strong protocol for 15 min (Fig. 1a) and a weak protocol for 6 min (Fig. 1d). The strong training produced an LTM that was successfully recalled after 1 day (Fig. 1c), and the weak training produced an STM that was recalled within 30 min, but not after 1 day (Fig. 1f). Lack of initial bias to one of the two objects during the training sessions was demonstrated through equal exploration preference to the two objects (Fig. 1b, e) and was confirmed for all groups.

### Optogenetic recall of an STM engram

Using the weak training protocol, we investigated whether STM has an engram, and which hippocampal pathway is involved in its processing, by performing optogenetic activation of the STM 1 day after its initial acquisition, a time point at which mice do not retrieve the memory (Fig. 1f). A previously established and validated system in labeling specific events in CA3 neurons was used[39], in which c-fos::tTA/KA1::Cre double transgenic mice were injected with AAV$_9$-TRE-DIO-ChR2-mCherry into their hippocampal CA3 to specifically label the activated CA3 cells involved in learning the weak event with the blue-light sensitive ChR2. One day after training, mice were subjected to the test session, during which optical stimulation to their CA3-CA1 projections was conducted by shining blue light (473 nm) above the CA1 area at 20 Hz stimulation (Fig. 2a–d). Mice that received this light stimulation (light ON group) explored the moved object significantly more than the unmoved object, compared to the control group (light OFF group) (Fig. 2e), which indicates the successful long-term recall of their STM. This result indicates that similar to the previously reported cases of retrograde amnesia and early-onset Alzheimer's disease[21,23], STM also forms an engram which can be artificially recalled 1 day later and that this engram is stored within the hippocampal CA3-CA1 circuit. However, when a different engram other than that of the NOL was labelled using the same labelling system (Supplementary Fig. 2a–c), mice explored the two objects equally during the artificial recall test session one day after the unlabelled NOL weak training (Supplementary Fig. 2d), reflecting the specificity of the aforementioned optogenetic recall of the labelled STM. Furthermore, the number of mCherry-positive cells was comparable between the group with the labelled NOL weak event and the group with the labelled circle (Supplementary Fig. 2e, f).

### Re-exposure facilitates STM consolidation

To test whether this revealed STM trace can have long-term effects, we tested the possibility that this trace can become activated by re-exposure to the same event in the future and whether this activation can lead to consolidation and natural long-term recall. Mice underwent the weak NOL training, and then, 1 day later, repeated the same training again and were then tested for consolidation after one more day (Fig. 3a–c). Mice subjected to this two-training paradigm succeeded to recall the STM 1 day after re-exposure, which indicates consolidation of the STM trace (Fig. 3d). This result

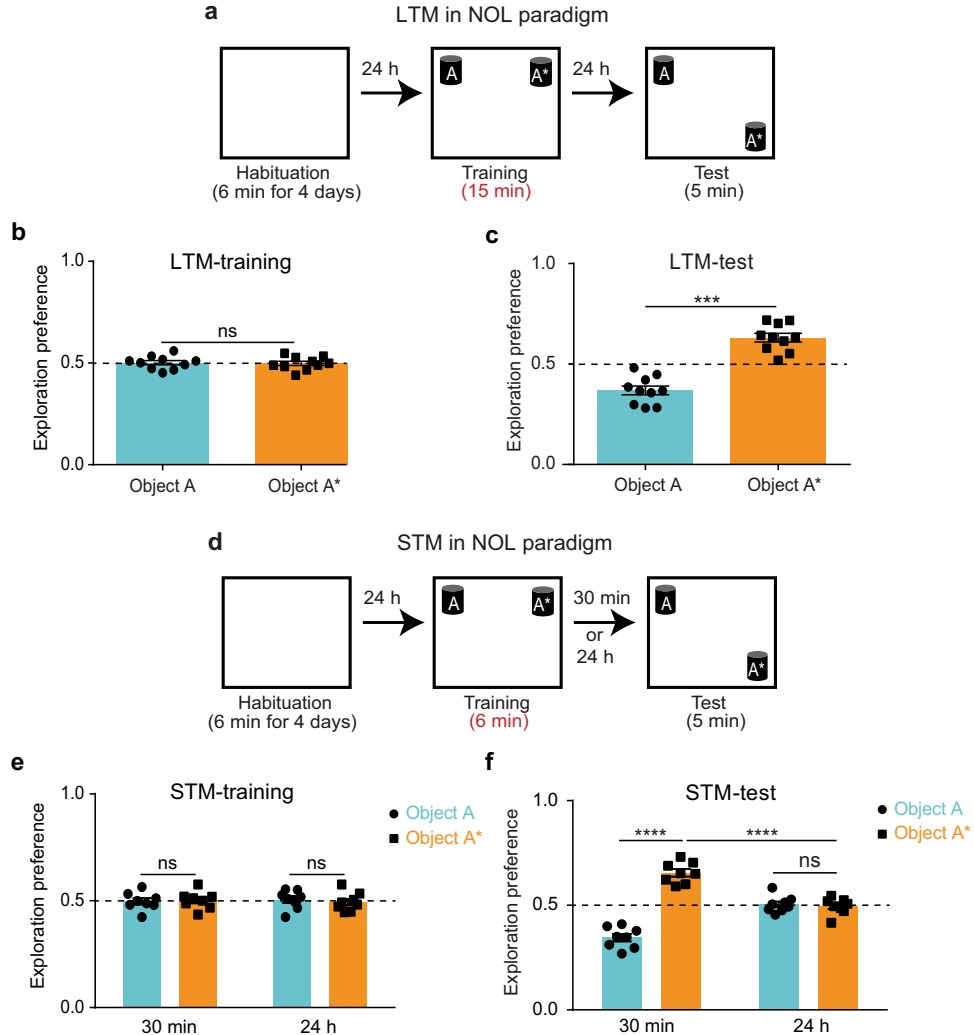

**Fig. 1 LTM and STM in novel object location task (NOL). a** NOL paradigm with the strong training protocol (15 min). Exploration preference for each object during the strong training (**b**) and the test sessions (**c**) (n = 10). **d** NOL paradigm with the weak training protocol (6 min). Exploration preference for each object during the weak training (**e**) and the test sessions (**f**) for groups tested after 30 min (n = 8) or 24 h (n = 8). Comparisons were made using paired student's t-test for comparing object A and object A* in the same group, and unpaired student's t-test for comparing object A* within different groups; ns not significant (P > 0.05), ***P < 0.001, ****P < 0.0001. Data are presented as the mean ± SEM.

suggests that the STM trace, produced by the first weak training, was used as a template by the second weak training to facilitate consolidation. Without the presence of the initial STM trace, the second weak event would have been processed independently and would have produced an STM, unable to be naturally retrieved one day later.

To assess how long this STM trace remains available for future re-usage, we implemented the two-training paradigm while increasing the spacing interval between the two training sessions (Fig. 3a–c). Using a 3-day interval, mice still recalled the memory similar to the 1-day interval; however, with a 6-day interval, the memory retrieval was no longer observed (Fig. 3d). These results indicate that the STM trace remains available for activation and subsequent consolidation if the same experience is repeated within 3 days.

**Long-term storage of STM trace is protein-synthesis dependent.** One of the major distinctions between STM and LTM is that LTM storage depends on new protein synthesis, but STM does not[25,26]. These new proteins are believed to alter the shape and function of the involved synapses which contributes to the

long retention time of LTM[5]. However, in case of STM, it is believed that new protein synthesis does not occur, but rather the covalent modification of pre-existing proteins causes temporary amplification of the synaptic transmission that contributes to the short retention time of STM[40].

Our finding that STM forms an engram made us curious as to whether STM triggers new proteins for its storage. We repeated the two-training paradigm with a 1-day interval and injected anisomycin, a protein synthesis inhibitor, to the hippocampal CA1 region immediately after the first training session (Fig. 4a–c and Supplementary Fig. 3). Mice injected with anisomycin did not recall the STM, unlike the PBS control group (Fig. 4d), which suggests that anisomycin disrupted the storage of the STM trace produced by the initial training, which led to failure in its consolidation after recurrence. This shows that while new protein synthesis is not involved in the same-day recall of STM, it is required for the long-term storage of its trace.

**Long-term storage and activation of STM trace is N-methyl-D-aspartate receptor-dependent.** NMDAR, the ionotropic gluta-mate receptor, plays a major role in memory processing where its

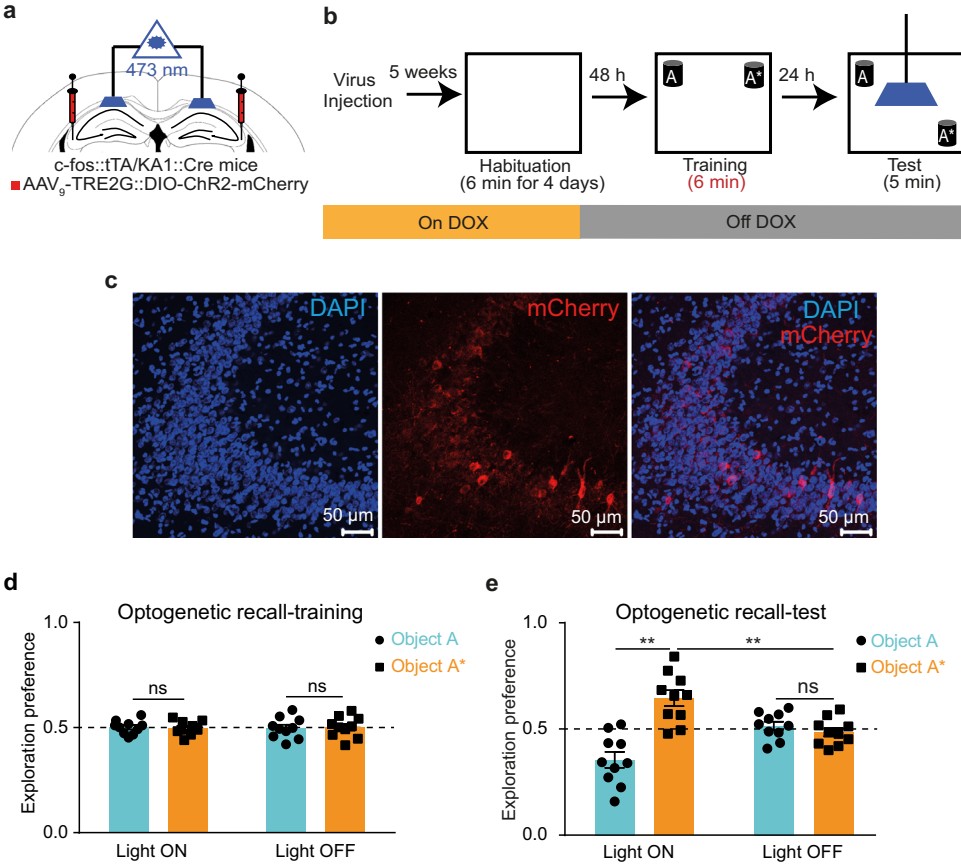

**Fig. 2 Optogenetic recall of an STM engram. a** Diagram of AAV injection into CA3, guide cannula and optic fiber placement into CA1. **b** Experimental design for labelling and optogenetic recall of the NOL STM engram. **c** Representative section of CA3 showing mCherry protein expression (scale bar = 50 μm, DAPI: 4',6-diamidino-2-phenylindole). Exploration preference for each object during the training (**d**) and the test (**e**) sessions in light ON ($n = 10$) vs. light OFF ($n = 10$) groups. Comparisons were made using paired student's t-test for comparing object A and object A* in the same group, and unpaired student's t-test for comparing object A* within different groups; ns, not significant ($P > 0.05$), **$P < 0.01$. Data are presented as the mean ± SEM.

activation leads to $Ca^{2+}$/calmodulin-dependent protein kinase II (CaMKII) phosphorylation, which in turn triggers a cascade of molecular events through which long-term potentiation (LTP) and synaptic plasticity occurs in the dendritic spines[5,27,28]; the storage site of specific memories[17]. D-AP5 is an NMDAR blocker that prevents the LTP of synaptic transmission and has been used to examine the role of NMDAR in learning and memory[41,42]. Similar to the anisomycin experiment, using the two-training paradigm with a 1-day interval, mice injected with AP5, in the hippocampal CA1 region after either the first or second weak trainings (Fig. 5a and Supplementary Figs. 4–6) explored the two objects equally in the test session compared with the PBS control groups (Fig. 5b, c) which indicates failure in the storage and activation of the STM trace, respectively. This further indicates that the NMDAR signaling pathway, and potentially synaptic plasticity, are involved in the facilitated consolidation of the STM by storing and activating the STM trace, consecutively.

**Causal link between offline activity during post-learning NREM sleep and long-term storage of STM trace.** Similar to protein synthesis and NMDAR activity; post-learning sleep has also been shown to be integral for the consolidation of long-term memories[29–34]. Accordingly, we tested the potential requirement for sleep in the long-term storage of the aforementioned STM trace.

To confirm the necessity of post-learning sleep in the long-term preservation of STM trace, we used the two-training paradigm with a 1-day interval in between, and subjected mice to a 5 h sleep deprivation immediately after the first training (Supplementary Fig. 7a–c). Contrary to the previous result (Fig. 3d), mice did not discriminate between the two objects during the test session (Supplementary Fig. 7d), which demonstrates a necessity for a post-learning sleep period to preserve the STM trace.

For a more specific approach targeting NREM sleep, a group of mice was injected with AAV$_9$-CaMKII-ArchT-eYFP (test group) and another group was injected with AAV$_1$-CaMKII-eYFP (control group). Orange light (589 nm) was delivered to their hippocampal CA1 region during all the NREM sleep stages that appeared within a 2 h sleep session immediately after the first training to optogenetically silence the offline hippocampal CA1 neurons (Fig. 6a–f; see "Methods" section for details). In line with the sleep deprivation results, ArchT-expressing mice failed to recall the memory compared with the eYFP-expressing control group (Fig. 6g) even though both groups had comparable duration of NREM sleep (Fig. 6h). Taken together, these results provide a causal link between post-learning offline activity, specifically during NREM sleep, and the long-term storage of STM trace.

## Discussion

Previous studies have extensively researched STM formation, the molecular pathways involved, and its distinctions from LTM[2–5]. However, the nature of the STM engram-wise and the fate of its trace beyond the few hours of its active recall remain unclear. Here, we demonstrate that the STM trace is not entirely erased

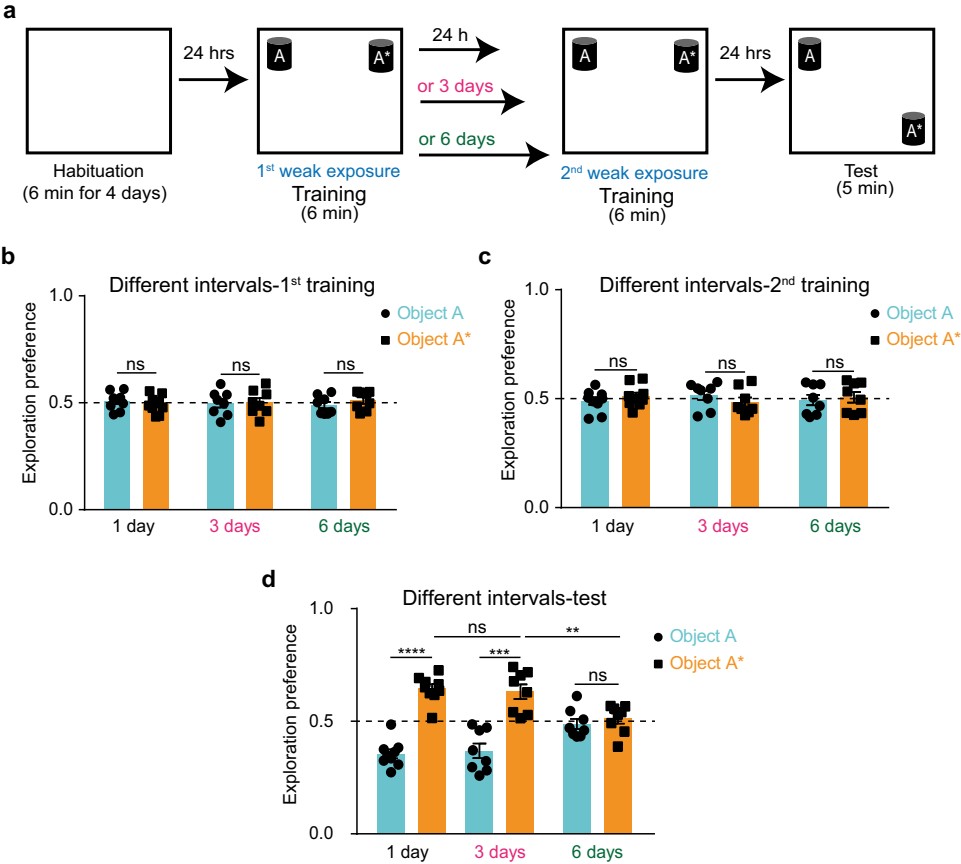

**Fig. 3 Re-exposure facilitates STM consolidation. a** NOL two-training paradigm with different intervals in between. Exploration preference for each object during the first training (**b**), the second training (**c**), and the test (**d**) sessions for different intervals groups; 1-day ($n = 9$), 3-day ($n = 8$), and 6-day ($n = 8$). Comparisons were made using two-way analysis of variance (ANOVA) followed by Bonferroni Post hoc test; ns not significant ($P > 0.05$), **$P < 0.01$, ***$P < 0.001$, ****$P < 0.0001$. Data are presented as the mean ± SEM.

from the brain within a few hours, but continues to exist as an engram even though it cannot be naturally retrieved. This finding redefines the STM and puts it in a similar context with other types of memories, initially thought to have been forgotten, yet with recent advances and technologies it became clear that traces of them continue to exist. This includes several cases of adult amnesia such as retrograde amnesia[21], early-onset Alzheimer's disease[23], and several cases of early infantile amnesia[43–45]. Furthermore, we found that this STM trace becomes consolidated when the same experience is encountered within 1 to 3 days, a finding that provides a possibility of converting an STM to an LTM; parallel to the well-established synaptic and behavioral tagging phenomena[14,38,46], by re-exposure to the same experience. Taken together, these results suggest that most likely our brain keeps a record of day-to-day weak events as an inactive template that primes the consolidation or strengthening of similar or reoccurring experiences in the near future.

Classical studies have shown that protein synthesis is not needed for the recall of novel weak experiences; however, these studies only focused on active STM recall hours from encoding[25,26,47]. We showed that STM does in fact trigger new protein synthesis crucial for the long-term storage and future activation of its trace. This result also provides the missing link between our finding that STM forms an engram and the classical notion that it does not trigger new protein synthesis. Thus, our results complement previous work by indicating that a weak event triggers new protein synthesis that is dispensable for active recall of the same memory on the same day, but crucial for the temporary storage of this event in a reversible trace that facilitates

future consolidation in the case of reoccurrence. However, future studies are needed to identify the proteins recruited in the long-term storage of STM trace and whether they are different from those recruited in LTM consolidation. Also, a correlation between the duration of the STM trace availability and the half-lives of these proteins remains to be established.

Our finding that NMDAR blockade affects both the storage and activation of STM trace suggests that a two-step process of synaptic plasticity may occur: a first step to store STM as an engram, and a second step to transform the STM engram into an active state. This suggestion is in line with previous results that engrams, which are unable to be naturally retrieved, have less spine density compared with active engrams[19,23]. However, further studies are needed to better understand the role of synaptic plasticity and to identify the downstream molecular players involved in this transformation.

Previous work has highlighted the necessity of post-learning sleep in memory consolidation, a process after which the engram can be actively and naturally recalled[30,35,36]. In line with this, we found that the STM trace is also stored after a period of post-learning sleep, whereby both generalized sleep deprivation and specified neuronal silencing during NREM disrupted the availability of this trace for future consolidation. Furthermore, a cross link has been shown to exist between sleep and protein synthesis where sleep deprivation has been shown to impair hippocampal protein synthesis and consequently memory consolidation[48–52]. This provides a direct connection between our finding that both post-learning sleep and post-learning protein synthesis are essential in the long-term storage of STM, with the former acting

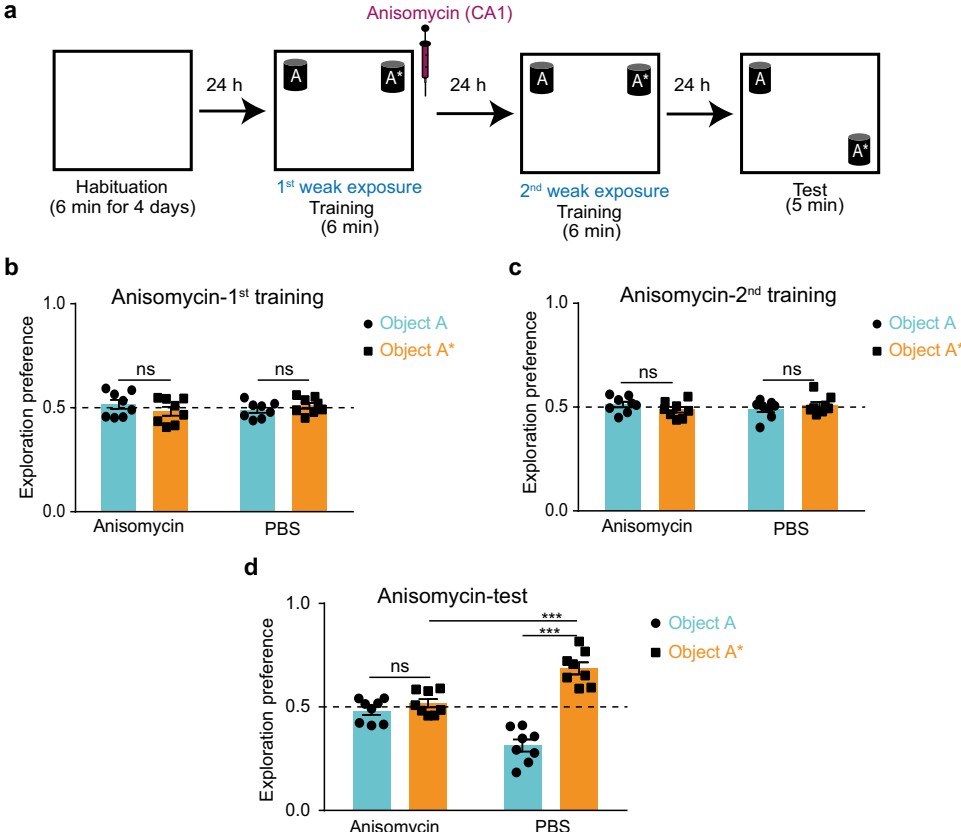

**Fig. 4 Long-term storage of STM trace is protein-synthesis dependent. a** NOL two-training paradigm with 1-day interval in between with anisomycin injection after the first training session into the CA1 region. Exploration preference for each object during the first (**b**) and the second (**c**) training sessions for anisomycin- ($n = 8$) and PBS- ($n = 8$) injected groups. **d** Exploration preference for each object during the test session for anisomycin- ($n = 8$) and PBS- ($n = 8$) injected groups. Comparisons were made using paired student's t-test for comparing object A and object A* in the same group, and unpaired student's t-test for comparing object A* within different groups; ns, not significant ($P > 0.05$), ***$P < 0.001$. Data are presented as the mean ± SEM.

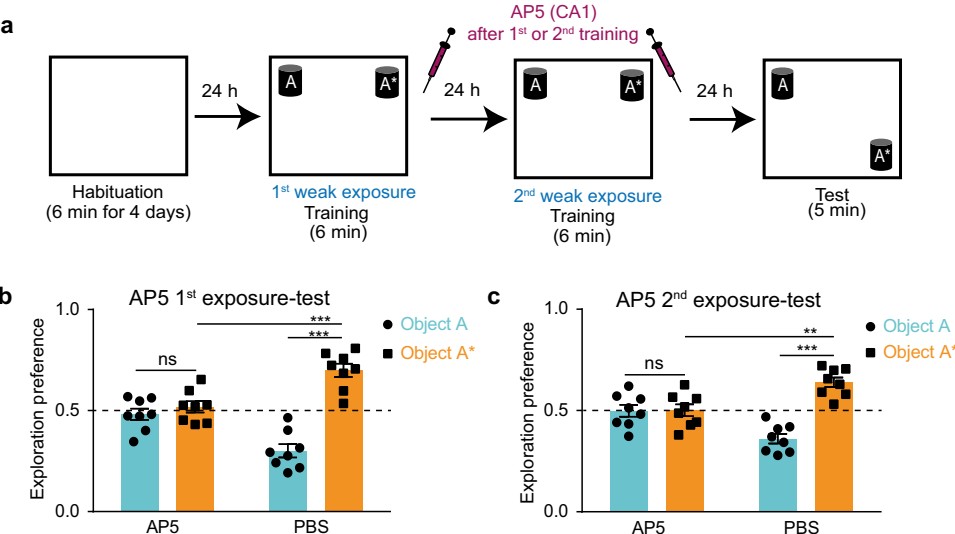

**Fig. 5 Long-term storage and activation of STM trace is N-methyl-D-aspartate receptor-dependent. a** NOL two-training paradigm with 1-day interval in between with AP5 injection after either the first or the second training sessions into the CA1 region. Exploration preference for each object during the test session for groups injected after the first (**b**) or the second (**c**) training sessions for AP5- ($n = 8$) and PBS- ($n = 8$) injected groups. Comparisons were made using paired student's t-test for comparing object A and object A* in the same group, and unpaired student's t-test for comparing object A* within different groups; ns, not significant ($P > 0.05$), **$P < 0.01$, ***$P < 0.001$. Data are presented as the mean ± SEM.

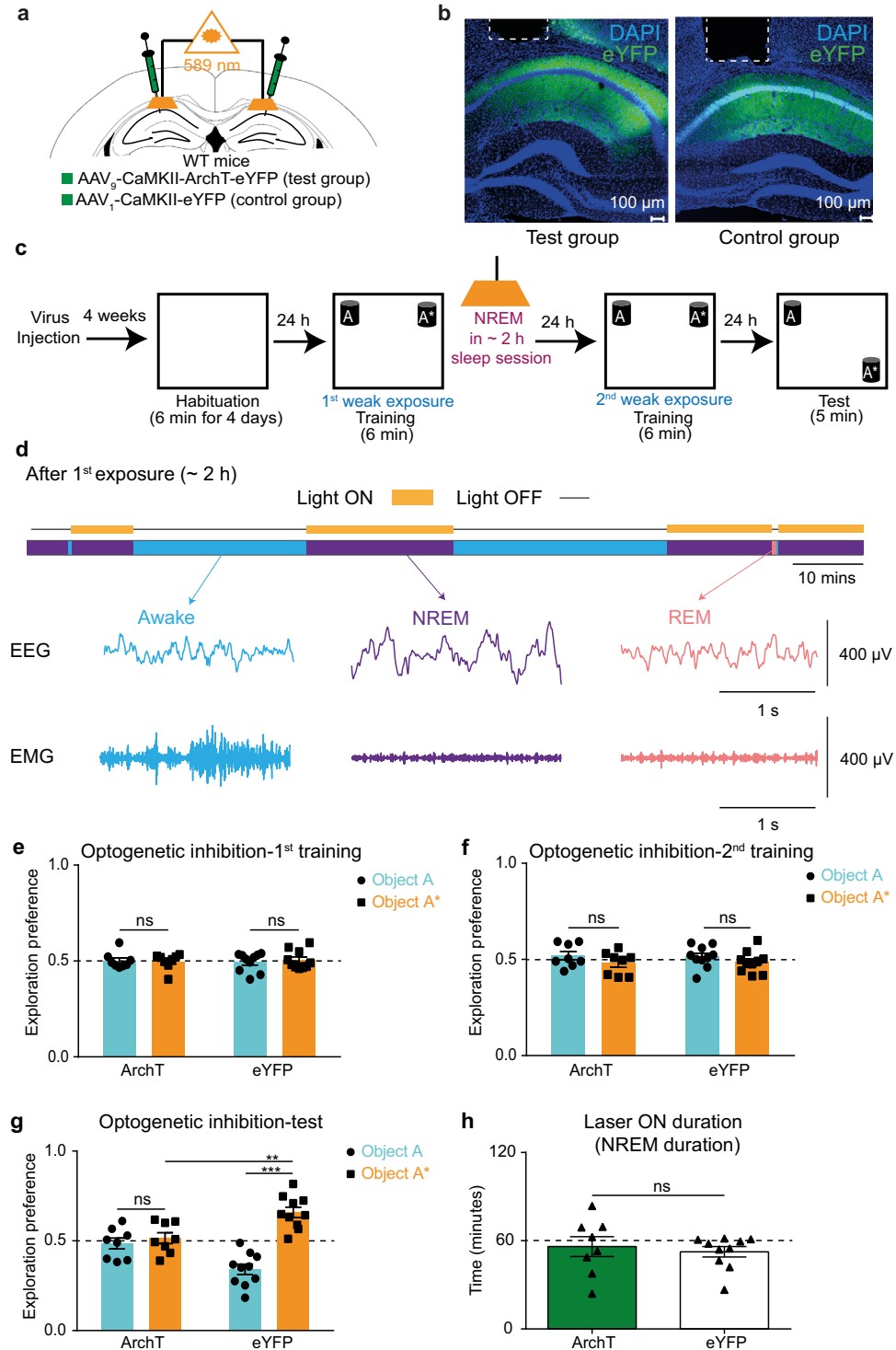

**Fig. 6 Causal link between offline activity during post-learning NREM sleep and long-term storage of STM trace. a** Diagram of AAV injection and CA1 optic fiber placement. **b** Representative CA1 sections showing ArchT-eYFP (Left: test group) and eYFP (Right: control group) protein expressions (scale bar = 100 µm; dashed line: guide cannula location). **c** Experimental design for optogenetic silencing of STM trace storage using NOL two-training paradigm during post-learning NREM sleep. **d** Example from one mouse showing 2-hour EEG/EMG recordings with laser ON during NREM state only (top), and example of EEG/EMG traces for NREM, REM, and awake states (bottom) (see Methods section for details). Exploration preference for each object during the first (**e**) and the second (**f**) training sessions for ArchT ($n = 8$) and eYFP ($n = 10$) groups. **g** Exploration preference for each object during the test session in ArchT ($n = 8$) and eYFP ($n = 10$) groups. **h** Duration of NREM sleep (laser ON) in ArchT ($n = 8$) and eYFP ($n = 10$) groups. Comparisons were made using wilcoxon signed rank sum test (**e**) and paired student's t-test for comparing object A and object A* in the same group, and unpaired Student's t-test for comparing object A* within different groups (**f**, **g**) and wilcoxon rank sum test (**h**); ns not significant ($P > 0.05$), **$P < 0.01$, ***$P < 0.001$. Data are presented as the mean ± SEM.

as a pre-requisite for the later. Another cross link exists between sleep and NMDAR activity where a period of post-learning sleep has been shown to be essential for synaptic plasticity[53,54] and altering NMDAR activation in return affects sleep[55,56]. Taken together, our results shed light on the crucial role of NREM sleep in the long-term storage of STM trace. However, given the important roles played by REM sleep and quiet awake states in LTM consolidation[36,57,58], similar to NREM sleep, other states may also be involved in the long-term storage of STM trace and its facilitated consolidation. NREM sleep may not be the exclusive state during which the STM trace is stored; therefore future studies are needed to investigate the potential roles played by other sleep or awake states and their mechanistic underpinnings.

Given the diversity of STM-inducing behavioral paradigms other than the NOL and the non-hippocampal forms of STM stored in many cortical regions, especially those that are part of the sensorimotor processing[59,60] or those that are part of planning[61], future studies are needed to elucidate whether other types of STM also have long-term traces regardless of their origin or function in the brain.

Finally, based on our results, we propose a model for STM consolidation in a two-step process which follows the initial-active state responsible for same day recall (Supplementary Fig. 8). First, STM is stored as an engram through post-learning NMDAR activity, new protein synthesis, and offline hippocampal neuronal activity during NREM. Second, STM engram is consolidated into an active engram by re-exposure within a few days of the initial event through NMDAR activity as well.

## Methods

**Animals**. Male c-fos::tTA/KA1::Cre double transgenic mice[39] were used in the optogenetic recall experiment. Male C57BL/6J mice were used in the optogenetic inhibition experiment. Male KA1::Cre mice[39] were used for other behavioral and pharmacological experiments. Mice were maintained on a 12 h light-dark cycle at 24 ± 3 °C and 55 ± 5% humidity with ad libitum access to food and water. All mice were aged 12–20 weeks at the time of behavioral experiments. All procedures involving the use of animals complied with the guidelines of the National Institutes of Health and were approved by the Animal Care and Use Committee of the University of Toyama (Approval numbers: A2019MED-35, A2022MED-7), Toyama, Japan.

**The novel object location task**. The NOL memory test was performed as described previously[23] with some modifications. Mice were subjected to 6 min handling and 6 min habituation, during which they freely explored an empty arena without objects for 4 consecutive days. Mice were then placed in the arena with two identical objects positioned in two different corners adjacent to the walls. Mice freely explored the objects for 15 min in the case of strong training or 6 min in the case of weak training. In the two-training protocol, mice were subjected to a second training session that was identical to the first training session in terms of context, object locations, and duration of exposure.

The test was conducted either after 30 min for short-term retention or 24 h for long-term retention[38]. Mice explored the arena for 5 min, and one object had been moved to a different corner and the other object remained unmoved. A video tracking system (Muromachi Kikai, Tokyo, Japan) was used to record exploration behavior for later analysis.

The arena was a square, previously used in a similar task[38]; one wall was a transparent acrylic board covered with a white tape, and the three other walls were grey acrylic (width 29 cm, depth 25 cm, height 29 cm). The floor consisted of grey acrylic board covered with white hard plastic. The objects used were colored ceramic cups placed upside down (width 7.8 cm, depth 5.5 cm, height 12.5 cm). The position of the two objects was counterbalanced across mice within each group. The arena and the objects were thoroughly cleaned with 80% ethanol between mice.

For exploration analysis, the time spent exploring each object was manually counted and the exploration preference for each object was calculated as follows[38]:

$$\text{Exploration preference} = \frac{\text{Time spent exploring each object}}{\text{Total time exploring both objects}} \quad (1)$$

A greater exploration preference for the moved object than for the unmoved object was considered to indicate that mice had the ability to recall the object locations from the training session. Exploration preference was also calculated for the training sessions; mice that showed any bias towards either object (an exploration preference > 0.6) were excluded from the dataset ($n = 7$). Exploration

was defined as sniffing or touching objects with the nose. Touching by hands, jumping on, or moving around the objects were not considered as exploratory behaviors[38].

**Sleep detection**. Electroencephalography (EEG) and electromyography (EMG) electrode placement and online sleep detection were performed as described previously[62] with modifications. For surgery, a custom-made 5-pin system that consisted of three wires terminating with screws was used; one screw was implanted in the parietal cortex for EEG recording, a ground screw and a reference screw were implanted in the cerebellar cortex, and two flexible wire cables were implanted in the neck muscle for EMG recording. All screws were fixed in place using dental cement. EEG and EMG signals were confirmed for each mouse individually before the start of experiments. All EEG/EMG recordings were performed using OpenEx Software Suite (Tucker Davis Technologies, USA) through a custom-made circuit file. EEG and EMG signals were amplified and filtered at 1–40 Hz for EEG and 65–150 Hz for EMG, and digitized at a sampling rate of 508.6 Hz.

**Online sleep state differentiation**. A custom-made circuit file was used to determine the sleep states by calculating the EMG root mean square (RMS) value, EEG delta power (1–4 Hz) RMS, EEG theta power (6–9 Hz), and RMS for 4 s epochs. The criteria for defining a sleep state were as follows. Awake: an EMG RMS that was greater than the threshold value determined for each mouse. NREM: a delta/theta ratio that was greater than 1. REM: a delta/theta ratio that was less than 1. Sleep state was defined if that state remained unchanged for three consecutive epochs (12 s). The sleep state defined by the program was confirmed by the experimenter through visual inspection of online waves and mouse activity, as follows. Awake: mouse is moving or remains still + low amplitude and high frequency (8–30 Hz) EEG + high EMG activity. NREM: mouse remains still + high amplitude and low frequency (0.5–4 Hz) EEG + low EMG activity. REM: mouse remains still + low amplitude and high frequency (4–9 Hz) EEG + flat EMG activity; a tremor may occasionally appear.

**Sleep deprivation**. Sleep deprivation was conducted as previously described[49] for the minimum induction of stress. Immediately after training, mice were returned to their home cages and the experimenter observed them for 5 consecutive hours. Whenever mice began to stand still or sleep, the experimenter gently tapped or pushed the home cage which alerted the mice and stopped them from sleeping.

**Pharmacological experiments**. Anisomycin (Sigma A9789, 62.5 µg/µl), a protein synthesis inhibitor, was dissolved with 1 N HCl and adjusted to pH 7.4 with NaOH[38]. D-AP5 (Tocris Bioscience, 0106, 30 mM), an N-methyl-D-aspartate receptor blocker, was dissolved in phosphate buffered saline (PBS) (T900, Takara BIO Inc, Japan)[13]. Drugs were aliquoted into single-use tubes and stored at −20 °C until use.

Mice (12 weeks) were anesthetized with a combination of three drugs (0.75 mg/kg, I.P., medetomidine (Domitor; Nippon Zenyaku Kogyo Co., Ltd., Japan), 4.0 mg/kg, I.P., midazolam (Fuji Pharma Co., Ltd., Japan), and 5.0 mg/kg, I.P., butorphanol (Vetorphale, Meiji Seika Pharma Co., Ltd., Japan)[63]. Mice were then placed on a stereotactic apparatus (Narishige, Japan), guide cannulas (C313GS-5/SPC, gauge 22, Plastics One, USA) were implanted bilaterally into the hippocampal CA1 region (AP −2.0 mm, ML ± 1.4 mm, DV 1.0 mm from bregma), and then dummy cannulas (C313IDCS-5/SPC, zero projection, Plastics One, USA) were inserted into the guide cannulas to protect them from dust. Micro screws were fixed near the bregma and lambda, and the guide cannulas were fixed in position using dental cement. Mice had a recovery period of 4 weeks in individual home cages before the start of the behavioral experiments.

Immediately after training, mice were anesthetized with isoflurane (099-06571, FUJIFILM Wako Pure Chemical, Osaka, Japan), and injection cannulas (C313IS-5/SPC, Plastics One, USA) were placed into guide cannulas projecting 0.5 mm below the tip of the guide cannulas. Mice were then returned back to their home cages. The internal cannulas were attached through a thin plastic tube filled with water to 10 µl Hamilton syringes that were fixed to an automated pump to maintain the drug flow rate at 0.2 µl/min. A total volume of 1 µl (anisomycin or D-AP5 as drugs, and PBS as a control) was injected bilaterally into the CA1 region, and the injection cannulas were left in place for 5 min after infusion to allow for drug diffusion. Mice were perfused 1 day after the test and the brain was extracted for histological examination.

**Viral vectors**. For the optogenetic recall experiment, AAV₉-TRE2G::DIO-ChR2(T159C)-mCherry (Titer: 1.3 × 10¹³ vg/ml)[39] was used. For the optogenetic inhibition experiment, AAV₉-CaMKII-ArchT-eYFP (Titer: 3.02 × 10¹⁶ vg/ml, a gift from Dr. R. Okubo-Suzuki) was used. The recombinant AAV₉ production was performed using a minimal purification method, and viral genomic titer was subsequently calculated as described previously[64]. Briefly, pAAV recombinant vector was produced using HEK293 T-cells (AAV293; 240073, Agilent Tech, CA, USA) cultured in 15 cm dishes (Corning, NY, USA). Cultured cells were maintained in Dulbecco's Modified Eagle Medium (D-MEM) (11995-065, GIBCO Life Technologies, USA) supplemented with 10% fetal bovine serum (FBS) (10270106,

GIBCO Life Technologies, USA), 1% 2 mM L-Glutamine (25030-149, GIBCO Life Technologies, USA), 1% 10 mM non-essential amino acid (MEM NEAA 100×, 11140-050, GIBCO Life Technologies, USA), and 1% (100×) penicillin-streptomycin solution (15140-148, GIBCO Life Technologies, USA). Confluent (70%) HEK293 T-cells were transfected using medium containing the constructed expression vector, pRep/Cap, and pHelper (240071, Agilent Technologies, Santa Clara, CA, USA) mixed with the transfection reagent polyethylenimine hydrochloride (PEI Max, 24765-1, Polysciences, Inc., Warrington, PA, USA) at a 1:2 ratio (W/V). After 24 h, the transfection medium was discarded, and cells were incubated for another 5 days in an FBS-free maintenance medium. On day 6, the AAV-containing medium was collected and purified from cell debris using a 0.45 µm Millex-HV syringe filter (SLHV033RS, Merck Millipore, Germany). The filtered medium was concentrated and diluted with D-PBS (14190-144, GIBCO Life Technologies, USA) twice using the Vivaspin 20 column (VS2041, Sartorius, Germany) after blocking the column membrane with 1% bovine serum albumin (01862-87, Nacalai Tesque, Inc., Japan) in PBS. To further calculate the titer, degradation of any residual cDNA in the viral solution from production was first assured by benzonase nuclease treatment (70746, Merck Millipore, Germany). Subsequently, viral genomic DNA was obtained after digestion with proteinase K (162-22751, FUJIFILM Wako Pure Chemical, Osaka, Japan), extraction with phenol/chloroform/isoamyl alcohol 25:24:1 v/v, and then precipitation with iso-propanol and final dissolution in TE buffer (10 mM Tris [pH 8.0], 1 mM EDTA). Titer quantification for each viral solution, referenced to that of the corresponding expression plasmid, was done using real-time quantitative PCR using THUN-DERBIRD SYBR qPCR Master Mix (QRS-201, Toyobo Co., Ltd, Japan) with the primers 5′-GGAACCCCTAGTGATGGAGTT-3′ and 5′-CGGCCTCAGT-GAGCGA-3′ targeting the inverted terminal repeat (ITR) sequence. The cycling parameters were adjusted as follows: initial denaturation at 95 °C for 60 sec, followed by 40 cycles at 95 °C for 15 s and at 60 °C for 30 s. For the control experiment, $AAV_1$-CaMKII-eYFP (titer $1.9 \times 10^{13}$ vg/ml, Addgene, #105622) was used.

**Stereotactic surgery**. Mice (12 weeks) were anesthetized and placed on a stereotactic apparatus as mentioned in the pharmacological experiments AAV (0.5 µl) was injected bilaterally into the hippocampal CA3 region (AP −2.0 mm, ML ± 2.3 mm from bregma, DV 2.0 mm from the dura) or the hippocampal CA1 region (AP −2.0 mm, ML ± 1.4 mm, DV 1.5 mm from bregma) using a glass micropipette filled with mineral oil attached to a 10 µl Hamilton syringe. The flow rate was fixed at 0.1 µl/min using a microsyringe pump and its automated controller (Narishige, Tokyo, Japan). The glass micropipette was left in place for 5 min after virus injection. Guide cannulas (C316GS-5/SPC, gauge 24, Plastics One, USA) were implanted bilaterally into the hippocampal CA1 region (AP −2.0 mm, ML ± 1.4 mm, DV 0.5 mm from bregma) and were covered with dummy cannulas (C316IDCS-5/SPC, zero projection, Plastics One, USA) to protect them from dust. Micro screws were fixed near the bregma and lambda, and the guide cannulas inserted were fixed using dental cement. Mice were maintained in individual home cages on 40 mg/kg Dox (doxycycline) for c-fos::tTA/KA1::Cre double transgenic mice or normal food pellets for C57BL/6J mice, and allowed to recover for 4–5 weeks before the start of behavioral experiments.

**Photo-stimulation during the test**. Mice in light ON and light OFF groups (Fig. 2) were maintained on Dox (40 mg/kg) during the handling and habituation sessions, and then were taken off Dox 48 h before the training session and remained off Dox until sacrificed. Mice with the non-NOL labelled engram (Supplementary Fig.2) were maintained on Dox (40 mg/kg) during the handling and habituation sessions, and then were taken off Dox 48 h before the labelling session of a circular context and then immediately returned to DOX (1000 mg/kg) after the labelling session and remained on DOX (1000 mg/kg) until sacrificed. On the test day, mice were anesthetized with isoflurane, and internal cannulas were replaced by a two-branch optical fiber unit consisting of a plastic cannula body and a 0.25 mm diameter optic fiber (COME2-DF2-250; Lucir, Ibaraki, Japan), which were placed inside the guide cannulas such that the tip of the optical fiber was targeted slightly above the CA1 region (DV 1.0 mm from bregma). Mice were returned to their home cages for 1 hour to recover from the anesthesia, and then moved to the experimental room where the fiber unit was fixed to an optical swivel above the test context (COME2-UFC; Lucir) that was connected to a laser source (200 mW, 473 nm, COME-LB473/200; Lucir). Pulses were delivered during the test session (5 min) according to a pre-fixed schedule using a stimulator (COME2-SPG-2; Lucir) in a time-lapse mode. Mice received 20 pulses of 473 nm blue light every second (20 Hz); each pulse had a duration of 0.5 ms and an inter-pulse interval of 49.5 msec. Mice were perfused 90 min after the test and their brains were extracted for histological examination.

**Photo-inhibition during sleep**. Immediately after the first training, mice were anesthetized with isoflurane and optic fibers were attached, as explained above, and then an EEG/EMG recording unit was attached. Mice were then placed into their home cages to sleep. For a duration of 2 h from sleep onset, 589 nm orange light was delivered above the hippocampal CA1 region, as explained above, whenever the mouse entered the NREM sleep stage. Mice were perfused 1 day after the test and their brains were extracted for histological examination.

**Histology for virus-infected animals**. Mice were deeply anesthetized with an overdose of pentobarbital solution and transcardially perfused using 4% paraformaldehyde in PBS, pH 7.4. The brains were then removed and post-fixed by immersing in 4% paraformaldehyde in PBS for 24 h at 4 °C, equilibrated in 25% sucrose in PBS for 2 days, and then frozen in dry-ice powder and stored at −80 °C until sectioning. Coronal sections of a 50 µm thickness were cut using a cryostat and then transferred to 12-well cell culture plates with 5 sections/well (Corning, Corning, NY) containing PBS solution. The floating sections were then treated with 4,6-diamidino-2-phenylindole (DAPI, 1 µg/ml, 10236276001; Roche Diagnostics) at room temperature for 20 min, and then washed 3 times with PBS (3 min/wash). The DAPI-stained sections were then mounted on a glass slide with ProLong Gold antifade reagent (Invitrogen, Thermo Fisher Scientific, Waltham, MA). Immuno-histochemistry (Supplementary Fig. 2e) was performed as previously described[17,65]. Briefly, the brain sections were incubated at room temperature for 1 h with a blocking buffer (3% normal donkey serum; S30, Chemicon by EMD Millipore, Billerica, MA, USA) in PBS solution containing 0.2% Triton X-100 and 0.05% Tween 20 (PBST). After the incubation, the buffer was discarded and then rabbit DsRed anti-mCherry (632496, Takara BIO Inc, Japan) primary antibody (1:500) in blocking solution was added for further incubation at 4 °C for 24–36 h. At the end of the incubation period, the primary antibody was removed, and sections were washed with 0.2% PBST three times for 10 min each. After washing, sections were treated with a complementary secondary antibody (1:1000), donkey anti-rabbit IgG Alexa Fluor 488 (A-21206, Thermo Fisher Scientific, USA), in blocking buffer solution at room temperature for 2–3 h. Simultaneously, nuclear staining was performed by adding 1 µg/mL DAPI in the buffer solution. After incubation, treatment was terminated by discarding the solution followed by three 10 min (0.2% PBST) washes before finally mounting the sections on glass slides with ProLong Gold antifade reagent. mCherry(+) cells were averaged out of 3 replicates in each mouse ($n = 4$). Images of native mCherry (Fig. 2c), immuno-stained mCherry (Supplementary Fig. 2e), and native eYFP fluorescence (Fig. 6b) were acquired using a Zeiss LSM 780 confocal microscope with a Plan-Apochromat 20×, 10×, and 5× objective lens (Nikon, Japan). All acquisition parameters were kept constant within each magnification and for all images. Unstained coronal sections were stored at −20 °C in a cryoprotectant solution (25% glycerol, 30% ethylene glycol, 45% PBS) for further use if needed.

**Histology for drug-injected animals**. Mice were sacrificed by decapitation, and brains were rapidly extracted and immediately frozen in dry-ice powder and stored at −80 °C until sectioning. Coronal sections of 30 µm thickness were cut using a cryostat, mounted onto glass slides, and air dried, and then examined to confirm the injection trace in the hippocampal tissue. Images were acquired on a fluorescence microscope (BZ9000; Keyence, Osaka, Japan) with a Plan-Apochromat 4× objective lens (Nikon, Tokyo, Japan). Mice were excluded if the injection trace was not clear ($n = 4$).

**Statistics and reproducibility**. Statistical analysis was performed using GraphPad Prism 6 (GraphPad Software, Inc., USA). Comparisons of data between two groups were performed using paired or unpaired Student's t-test (two-tailed) and the Wilcoxon rank sum or signed rank sum tests for groups which did not show normal distribution (Fig. 6e, h). Multiple-group comparisons were assessed using a one-way or a two-way analysis of variance (ANOVA) followed by post-hoc Bon-ferroni multiple comparisons test. Quantitative data are expressed as the mean ± SEM. Detailed information on sample sizes and statistical analysis are provided in Supplementary Data 1: sampling and statistical analysis details.

**Reporting summary**. Further information on research design is available in the Nature Research Reporting Summary linked to this article.

## Data availability
All data needed to evaluate the conclusions in the paper are present in the paper and the Supplementary Materials: Supplementary Data 1 and Supplementary Data 2.

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

## Acknowledgements

We thank K. Deisseroth (Stanford University) for the ChR2 (T159C) AAV vector and the ArchT-eYFP plasmid, A. Konno and H. Hirai for disclosing the AAV₉ virus production protocol prior to publishing, and R. Okubo-Suzuki for preparation of the AAV₉-

CaMKII-ArchT-eYFP viral vector. We thank M.H. Aly and K. Nomoto for technical assistance, S. Tsujimura and S. Okami for maintaining mice, K. Choko and Y. Saitoh for preparation of EEG/EMG electrodes, N. Oishi for technical teaching, and all members of the Inokuchi Laboratory for discussions and suggestions. This work was supported by JSPS KAKENHI (JP18H05213), the Core Research for Evolutional Science and Technology (CREST) program (JPMJCR13W1) of the Japan Science and Technology Agency (JST), a Grant-in-Aid for Scientific Research on Innovative Areas "Memory dynamism" (JP25115002) from MEXT, and the Takeda Science Foundation (to K.I.). Additional support was provided by JSPS KAKENHI (20H03554 and 17K19445), THE HOKURIKU BANK Grant-in-Aid for young scientists, THE FIRSTBANK OF TOYAMA Scholarship Foundation research grant, and the Takeda Science Foundation (to M.N.). Grant-in-Aid for young scientists from JSPS KAKENHI (JP 19K16892) (to K.A.). The Otsuka Toshimi Scholarship Foundation supported M.W.

## Author contributions

M.W., M.N., K.A., and K.I. designed the experiments and wrote the manuscript. M.W. performed all experiments except Supplementary Fig. 2. M.N. and E.M. performed experiment in Supplementary Fig. 2. M.W. and M.N. analysed the data. K.I. supervised the entire project.

## Competing interests

The authors declare no competing interests.
