## [Peer Review File · Communications Biology]

Reviewers' comments:

Reviewer #1 (Remarks to the Author):

This is an important and novel study, which shows that short-term memory traces do not decay over time, and in fact are retained as persistent engrams.

My major experimental criticism is the lack of an "information specificity control". For Figure 2, the authors need to show that activation of a random sparse population, or a different engram, does not induce the same optogenetic effect. This is essential for the conclusions of the paper to be robust.

Figure 6 seems solid overall, but needs to be reviewed closely by a calcium imaging expert.

Figure 7 seems quite out of place, and needs to be better justified in the context of this study.

The term "silent engram" is inappropriate here. If the engram was silent, then it would not be accessible during the 2nd weak training session. For the 2nd weak training to have an effect, it must be able to access and update the first engram. So the first engram was perfectly available, but just not expressed behaviorally under those conditions.

Some of the Figures are on the grainy side, and the text should be carefully proof read.

Reviewer #2 (Remarks to the Author):

Reviewer comments:

Wally et al. study the possibility that STM ensembles exist as a salient form for longer time periods. They use a NOL task for STM and combine calcium imaging and optogenetics to show that STM is maintained in a silent form beyond several hours. This silent form may be consolidated by recurrence after 3 days. The manuscript is clearly written and figures are easy to understand. Most of the manuscript is simple and attractive to the naïve reader which is an enormous advantage (but see major point 1 for an exception). The results are novel and may lead to further insights in the field. I strongly feel that this manuscript is highly suitable for publication, but I still have several major concerns and some minor issues:

Major concerns:

1. Although the MS is clearly written, this comes into a disadvantage when getting to Figure 6 relating to the calcium imaging experiments. First, this part in the result section is not detailed enough and it was not clear from the text what is the method used here and what exactly did they do in terms of data analysis, which are both important to have in the results section because they give the reader an insight into what is quantified in this section. When using such complex systems as the miniscope, it is customary to show an example field of view of the imaged cells, so the reader could get an impression of the image quality. In addition, an example calcium trace from several neurons is also a standard in the field so the reader can assess the signal to noise ratio. These signals have a lot of noise which can be sometimes correlative to different task variables. Furthermore, there needs to be more info, preferably with a figure, into how the authors align the FOVs across the different periods, and how they select the neurons, ensuring that they are measuring from the same cells across the whole experiment. This is crucial in light of their conclusions of overlapping cells.

Next, the authors show a Mean Z-score of all neurons during each period with and without normalization to the habituation period (Fig. 6d, e). From figure 6d, especially without showing any time traces, it is hard to observe any activity at all since the z score values are low and vary across period, even in the same neuron. Is the signal above the noise level? Another missing issue is more specifics into what the mouse was actually doing? I guess the mouse was moving more in Tr1 compared to Habi. This can effect the higher activity in TR1. But I could not find a mention of whether the mice were recorded with an overview camera during these experiments. Another thought, if the activity reflects STM consolidation, then should activity be elevated specifically in the test period, when the mouse is near the new object location? Can the authors separate activity to different location in space in order to better link activity to STM reactivation? Shouldn't there be

an activity difference between tr1 and tr2 when the STM is reoccurred? The fact that activity levels are high across all periods for almost all neurons (except for the Habi period for a small amount of cells), makes me unconvinced about the results of this section. There are many open questions in this part of the study.

Second, the authors also add NREM sleep which was not tested in the previous sections. This complicates things, especially since activity levels are just as high as TR1,2 and test. Does this really mean that these neurons hold an STM trace? If the authors measure from mice as they just freely move in the cage, or during REM sleep, or just sitting quietly, do they see the same activity levels. If so, then it would be unlikely that activity levels are linked to a silent STM trace.

Moreover, the neurons measured from CA1 are non-specific and may not all belong to an STM engram. This is unlike figure 2 and may also average out significant results. In summary, this part of the manuscript is currently unconvincing and seems to be detached from the rest of the MS.

2. Another disadvantage of the clear and simple writing style is that the authors take the reader through a very clear line of thought, whereas the nature of STM is actually quite broad. For example, in their first sentence in the intro the authors write: "Short-term memory (STM) is formed by the delivery of a weak stimulus while long-term memory (LTM) is formed by a stronger and more enduring stimuli". This may be true, but this is not the only definition to STM. STM is the maintenance of memory for several second that enables to make available useful information. There is a whole line of studies, even using mice, that relate to STM in a different manner (e.g. Li, Svoboda et al. 2015; Gilad, Helmchen et al. 2018, Goard, Sur et al. 2016), where information of a stimulus is stored until it is used. In their next sentence, the authors write: "Moreover, STM is 29 known to last for only a few hours, whereas LTM is subjected to a consolidation process that 30 allows it to remain for long periods of time." This may be true in this specific paradigm, but in other paradigms STM is thought to be maintained for only several seconds. Is there a difference between different types of STMs? What is the link between this type of STM in CA1 and other types of STM found in the cortex? Due to its broad definition, it will be informative if the authors discuss how their results are linked in a broader sense and how it may be linked to other types of STM studies.

Minor issues:

1. The authors use a t-test throughout most of their analysis, but t-test assumes normality. The authors should either test each data set for normality or use non-parametric statistics.
2. In general, it would be nice to have a little bit more detail on the methods and data analysis in the results section.
3. How many mice were used for the calcium imaging experiments?
4. Can you give more details on what is the logic of injecting an AAV into CA3 and silencing CA1 (in Figure 2), other than just mentioning that this system was previously validated. In general, I would like to get more details on why CA1 was chosen to maintain STM as compared to other potential areas. And discuss other areas that may participate in the engram of STM (linked to major point 2)
5. The model in Supp figure 8 is not very clear for me. The graph requires more description. Also, once better explained, I think this would be very informative in the main results.

Response to reviewers

We extend our appreciation to the reviewers for their prompt reply and thorough revision, we are grateful for their feedback and constructive comments, and we greatly value the time and effort they spent on our manuscript and the careful remarks that helped us revise our manuscript and present it in the best possible shape. We have addressed all of the reviewers' comments and provide our point- by-point response to the comments below. We have revised our manuscript, guided by the reviewers' feedback, **and all corrections or additions made to the text are colored in red.**

We hope that our revised manuscript has addressed the reviewers' concerns and is now suitable for publication in *Communications Biology*.

Reviewer #1 (Remarks to the Author):

This is an important and novel study, which shows that short-term memory traces do not decay over time, and in fact are retained as persistent engrams.

→ We are grateful to the reviewer for their thorough reading and constructive feedback which helped us revise our manuscript to the best of our knowledge and present it to the scientific community the best way we can. We have taken into consideration all their comments and kind suggestions and revised our manuscript accordingly.

1) My major experimental criticism is the lack of an "information specificity control". For Figure 2, the authors need to show that activation of a random sparse population, or a different engram, does not induce the same optogenetic effect. This is essential for the conclusions of the paper to be robust.

→ We thank the reviewer for pointing out this control group. We have carried out this control experiment which is now presented as **Supplementary Fig.2**. Briefly, we used the same labelling system (AAV and transgenic mice), but instead of labelling the NOL weak training event, we labelled a different context (circle) then we exposed the mice to the NOL weak training. We closed the DOX window immediately after labelling the circle to make sure that the NOL event was not labelled. One day after the labelling session, we exposed the mice to the NOL weak training (without labelling) then we tested for STM artificial recall the next day using the same optogenetic stimulation protocol as previously used (20 Hz) (**Supplementary Fig. 2a-b**). As a result, mice did not artificially recall the NOL STM on the test day (**Supplementary Fig. 2c**), which confirms the specificity of our previous STM engram results.

→ We have also confirmed the Chr2 labelling using immunohistochemistry (**Supplementary Fig. 2d**) and confirmed by counting that the number of mCherry-positive cells is comparable between the group with labelled NOL and the control group with labelled circle (**Supplementary Fig. 2e**), which rules out that the behavioral result is due to a difference in the number of labelled neurons between the two labelling strategies.

New Supplementary Fig. 2

→ We have edited our results section as follows:

Line 113-119 However, when a different engram other than that of the NOL was labelled using the same labelling system (Supplementary Fig. 2a-b), mice explored the two objects equally during the artificial recall test session one day after the unlabelled NOL weak training (Supplementary Fig. 2c), reflecting the specificity of the aforementioned optogenetic recall of the labelled STM. Furthermore, the number of mCherry-positive cells was comparable between the group with the labelled NOL weak event and the group with the labelled circle (Supplementary Fig. 2d-e).

→ We have also edited our methods section to include the newly added control experiment in the optogenetic manipulation part and added the immunohistochemistry part.

Line 470-476 Mice in light ON and light OFF groups (Fig.2) were maintained on Dox (40 mg/kg) during the handling and habituation sessions, and then were taken off Dox 48 hours before the training session and remained off Dox until sacrificed. Mice with the non-NOL labelled engram (Supplementary Fig.2) were maintained on Dox (40 mg/kg) during the handling and habituation sessions, and then were taken off Dox 48 hours before the labelling session of a circular context and then immediately returned to DOX (1000 mg/kg) after the labelling session and remained on DOX (1000 mg/kg) until sacrificed

Line 506-520 Immunohistochemistry (Supplementary Fig. 2d) was performed as previously described^{17,53}. Briefly, the brain sections were incubated at room temperature for 1 h with a blocking buffer (3% normal donkey serum; S30, Chemicon by EMD Millipore, Billerica, MA, USA) in PBS solution containing 0.2% Triton X-100 and 0.05% Tween 20 (PBST). After the incubation, the buffer was discarded and then rabbit DsRed anti-mCherry (632496, Takara BIO Inc, Japan) primary antibody (1:500) in blocking solution was added for further incubation at 4°C for 24–36 h. At the end of the incubation period, the primary antibody was removed, and sections were washed with 0.2% PBST three times for 10 min each. After washing, sections were treated with a complementary secondary antibody (1:1000), donkey anti-rabbit IgG Alexa Fluor 488 (A-21206, Thermo Fisher Scientific, USA), in blocking buffer solution at room temperature for 2–3 h. Simultaneously, nuclear staining was performed by adding 1 µg/mL DAPI in the buffer solution. After incubation, treatment was terminated by discarding the solution followed by three 10-min (0.2% PBST) washes before finally mounting the sections on glass slides with ProLong Gold antifade reagent. mCherry(+) cells were averaged out of 3 replicates in each mouse (n=4).

2) Figure 6 seems solid overall, but needs to be reviewed closely by a calcium imaging expert.

→ We have carefully reviewed and considered reviewer#2's comments regarding the calcium imaging experiment and we agree with them that this experiment in its current format may weaken and average out our significant results. Accordingly, we have decided to omit this part from our revised manuscript which will give us more time in the future to revise the imaging strategy we implemented, choose a better method of analysis and hopefully represent our revised findings in a subsequent publication to this manuscript.

→ Accordingly, we have removed previous Fig. 6 and Supplementary Fig. 6 from our data.

3) Figure 7 seems quite out of place, and needs to be better justified in the context of this study.

→ From Figure 4 to the end, our main aim was to characterize the requirements for the storage of the silent STM trace. Accordingly, we tested the necessity of post-learning protein synthesis and post-learning NMDAR activity.

→ Similarly, in this figure (now Fig. 6) we aimed to test whether a period of post-training sleep is needed for the long-term storage of the silent STM trace as it is needed for the consolidation of LTM.

→ For this purpose, we carried out two experiments, one experiment using a general approach to block any kind of sleep after 1st STM training and so we did the sleep deprivation experiment (Supplementary Fig. 7) to examine whether post-learning sleep in general was involved in the long-term processing of the STM, without highlighting which sleep stage.

→ The second experiment was conducted using a more specified approach to optogenetically silence the hippocampal CA1 activity during NREM sleep at the same time point after the 1st STM

training in order to get a causal link between the necessity of sleep, specifically NREM sleep, on the long-term storage of the silent STM trace.

→ We believe that these experiments are justified in the context of our study and provide novel insight into the necessity of sleep for weak memory traces similar to strong memory traces as we explained in our discussion section:

Line 288-296 Previous work has highlighted the necessity of post-learning sleep in memory consolidation, a process after which the engram can be actively and naturally recalled¹⁻³. In line with this, we found that the silent STM trace is also stored after a period of post-learning sleep, whereby both generalized sleep deprivation and specified neuronal silencing during NREM disrupted the availability of this trace for future consolidation. Furthermore, sleep deprivation has been shown to impair hippocampal protein synthesis and consequently memory consolidation⁴. This provides a direct connection between our finding that both post-learning sleep and post-learning protein synthesis are essential in the long-term storage of STM, with the former acting as a pre-requisite for the later.

4) The term "silent engram" is inappropriate here. If the engram was silent, then it would not be accessible during the 2nd weak training session. For the 2nd weak training to have an effect, it must be able to access and update the first engram. So the first engram was perfectly available, but just not expressed behaviorally under those conditions.

→ We agree with the reviewer that this engram was “*perfectly available but just not expressed behaviorally under those conditions*”, hence the term “silent engram”. Because, according to the literature¹⁻³, the main criteria for defining an engram as “silent” is that this engram is not naturally recalled (not behaviorally expressed as mentioned by the reviewer) but can be recalled upon artificial stimulation.

→ In our case we have proven that these 2 conditions exist in our behavioral task where STM was not naturally recalled after 1 day (Fig. 1f) but could be artificially recalled by optogenetic stimulation at the same time point (Fig. 2d). And so we believe we have fulfilled the criteria agreed upon in literature to use the term “silent engram”.

→ We also believe that this finding is in agreement with our 2-training protocol results, where the 1st weak training produced a silent (not-naturally or behaviorally expressed) engram which was “*perfectly available*” like the reviewer mentions but in a silent form, that’s why the 2nd weak training was able to access it (even in its silent form) and activate it by turning it into a naturally and behaviorally expressed engram (active engram). So both our understanding and the reviewer’s opinion are in agreement regarding the availability of the first engram after the 1st weak training.

5) Some of the Figures are on the grainy side, and the text should be carefully proof read.

→ Some figures may have appeared on the grainy side in the initial submission documents due to the limitations on the figure resolution. For the revised manuscript, we have provided high resolution figures and will submit the actual .ai figures as well when requested by the editor.

→ The manuscript has been carefully read by all the co-authors and has been proof-read by an English editing service for grammatical and structural mistakes.

Reviewer #2 (Remarks to the Author):

Reviewer comments:

Wally et al. study the possibility that STM ensembles exist as a salient form for longer time periods. They use a NOL task for STM and combine calcium imaging and optogenetics to show that STM is maintained in a silent form beyond several hours. This silent form may be consolidated by recurrence after 3 days. The manuscript is clearly written and figures are easy to understand. Most of the manuscript is simple and attractive to the naïve reader which is an enormous advantage (but see major point 1 for an exception). The results are novel and may lead to further insights in the field. I strongly feel that this manuscript is highly suitable for publication, but I still have several major concerns and some minor issues:

→ We are grateful to the reviewer for thorough reading and constructive feedback which helped us revise our manuscript to the best of our knowledge and present it to the scientific community the best way we can. We have taken into consideration all their comments and kind suggestions and revised our manuscript accordingly.

Major concerns:

1) Although the MS is clearly written, this comes into a disadvantage when getting to Figure 6 relating to the calcium imaging experiments. First, this part in the result section is not detailed enough and it was not clear from the text what is the method used here and what exactly did they do in terms of data analysis, which are both important to have in the results section because they give the reader an insight into what is quantified in this section. When using such complex systems as the miniscope, it is customary to show an example field of view of the imaged cells, so the reader could get an impression of the image quality. In addition, an example calcium trace from several neurons is also a standard in the field so the reader can assess the signal to noise ratio. These signals have a lot of noise which can be sometimes correlative to different task variables. Furthermore, there needs to be more info, preferably with a figure, into how the authors align the FOVs across the different periods, and how they select the neurons, ensuring that they are measuring from the same cells across the whole experiment. This is crucial in light of their conclusions of overlapping cells. Next, the authors show a Mean Z-score of all neurons during each period with and without normalization to the habituation period (Fig. 6d, e). From figure 6d, especially without showing any time traces, it is hard to observe any activity at all since the z score values are low and vary across period, even in the same neuron. Is the signal above the noise level? Another missing issue is more specifics into what the mouse was actually doing? I guess the mouse was moving more in Tr1 compared to Habi. This can effect the higher activity in TR1. But I could not find a mention of whether the mice were recorded with an overview camera during these experiments. Another thought, if the activity reflects STM consolidation, then should activity be elevated specifically in the test period, when the mouse is near the new object location? Can the authors separate activity to different location in space in order to better link activity to STM reactivation? Shouldn't there be an activity difference between tr1 and tr2 when the STM is reoccurred? The fact that activity levels are high across all periods for almost all neurons (except for the Habi period for a small amount of cells), makes me unconvinced about the results of this section. There are many open questions in this part of the study.

Second, the authors also add NREM sleep which was not tested in the previous sections. This complicates things, especially since activity levels are just as high as TR1,2 and test. Does this

really mean that these neurons hold an STM trace? If the authors measure from mice as they just freely move in the cage, or during REM sleep, or just sitting quietly, do they see the same activity levels. If so, then it would be unlikely that activity levels are linked to a silent STM trace. Moreover, the neurons measured from CA1 are non-specific and may not all belong to an STM engram. This is unlike figure 2 and may also average out significant results.

In summary, this part of the manuscript is currently unconvincing and seems to be detached from the rest of the MS.

→ We thank the reviewer for the honest and thorough criticism to our calcium imaging experiment which made us re-question our strategy and results. We have carefully reviewed and considered all the comments and we agree that this experiment in its current format may weaken and average out our significant results. Accordingly, we have decided to omit this part from our revised manuscript which will give us more time in the future to revise the imaging strategy we implemented, choose a better method of analysis, and hopefully represent our revised findings in a subsequent publication to this manuscript.

→ Accordingly, we have removed previous Fig.6 and Supplementary Fig. 6 from our data.

2. Another disadvantage of the clear and simple writing style is that the authors take the reader through a very clear line of thought, whereas the nature of STM is actually quite broad. For example, in their first sentence in the intro the authors write: "Short-term memory (STM) is formed by the delivery of a weak stimulus while long-term memory (LTM) is formed by a stronger and more enduring stimuli". This may be true, but this is not the only definition to STM. STM is the maintenance of memory for several second that enables to make available useful information. There is a whole line of studies, even using mice, that relate to STM in a different manner (e.g. Li, Svoboda et al. 2015; Gilad, Helmchen et al. 2018, Goard, Sur et al. 2016), where information of a stimulus is stored until it is used. In their next sentence, the authors write: "Moreover, STM is 29 known to last for only a few hours, whereas LTM is subjected to a consolidation process that 30 allows it to remain for long periods of time." This may be true in this specific paradigm, but in other paradigms STM is thought to be maintained for only several seconds. Is there a difference between different types of STMs? What is the link between this type of STM in CA1 and other types of STM found in the cortex? Due to its broad definition, it will be informative if the authors discuss how their results are linked in a broader sense and how it may be linked to other types of STM studies.

→ We thank the reviewer for pointing out these studies and questions. We agree with the reviewer on the importance of raising these questions in our discussion.

→ We believe our results do not disagree with any of the mentioned studies as all these studies and others have focused on the same-day maintenance of the STM regardless of how long it lasts. However, in our study we focus on the potential long-term existence of the STM and provide evidence to the possibility of its existence beyond the time point addressed in other STM studies.

→ In our work, we have studied a hippocampal dependent type of STM (NOL behavioral paradigm) and so our manipulations have targeted the CA1 brain region. The studies mentioned by the reviewer mainly used cortical dependent type of STM as most of their studies were related to sensory processing. Whether our findings also apply to other types of STM, using other behavioral tasks or stored in other brain regions, remains to be investigated by future studies.

→ Accordingly, we have edited our discussion section to raise these questions and to better relate

our work to other STM studies as mentioned by the reviewer ⁴⁻⁶.

Line 303-307 Given the diversity of STM-inducing behavioral paradigms other than the NOL and the non-hippocampal forms of STM stored in many cortical regions, especially those that are part of the sensorimotor processing ^{47,48} or those that are part of planning ⁴⁹, future studies are needed to elucidate whether other types of STM also have a long-term silent trace regardless of their origin or function in the brain.

Minor issues:

1. The authors use a t-test throughout most of their analysis, but t-test assumes normality. The authors should either test each data set for normality or use non-parametric statistics.

→ We thank the reviewer for their request and accordingly we have performed the normality test D'Agostino & Pearson test for all the data sets in which we used the t-test.

→ All data sets have passed the normality test, so we continued to use t-test for analysis except data sets in (now Fig. 6e and Fig. 6h) and so we changed its analysis as requested by the reviewer from t-test to the non-parametric Wilcoxon rank sum and signed rank sum tests and got similar results to those previously reported.

→ Below is the summarized normality testing results for our data sets including the newly added control group in (now Supplementary Fig.2), highlighted are the data sets which did not pass the normality test.

Fig. #	Group		Normality test	K2 value	p-value	p-value summary	Passed normality test ? (alpha=0.05)
1b	Object A		D'Agostino & Pearson test	0.06359	0.9687	ns	yes
	Object A*			0.06359	0.9687	ns	yes
1c	Object A			0.4815	0.7861	ns	yes
	Object A*			0.4815	0.7861	ns	yes
1e	30 minutes	Object A		0.812	0.6663	ns	yes
		Object A*		0.812	0.6663	ns	yes
	24 hours	Object A		1.615	0.4459	ns	yes
		Object A*		1.615	0.4459	ns	yes
1f	30 minutes	Object A		0.7429	0.6897	ns	yes
		Object A*		0.7429	0.6897	ns	yes
	24 hours	Object A		3.605	0.1649	ns	yes
		Object A*		3.605	0.1649	ns	yes
2c	Light ON	Object A		0.1078	0.9475	ns	yes
		Object A*		0.1078	0.9475	ns	yes
	Light OFF	Object A		0.2967	0.8621	ns	yes
		Object A*		0.2967	0.8621	ns	yes
2d	Light ON	Object A	0.3376	0.8447	ns	yes	
		Object A*	0.3376	0.8447	ns	yes	
	Light OFF	Object A	0.5831	0.7471	ns	yes	
		Object A*	0.5831	0.7471	ns	yes	
4b	Anisomycin		Object A	3.558	0.1688	ns	yes

		Object A*	3.558	0.1688	ns	yes
	PBS	Object A	0.7108	0.7009	ns	yes
		Object A*	0.7108	0.7009	ns	yes
4c	Anisomycin	Object A	0.346	0.8412	ns	yes
		Object A*	0.346	0.8412	ns	yes
	PBS	Object A	5.444	0.0657	ns	yes
		Object A*	5.444	0.0657	ns	yes
4d	Anisomycin	Object A	3.537	0.1706	ns	yes
		Object A*	3.537	0.1706	ns	yes
	PBS	Object A	0.6148	0.7354	ns	yes
		Object A*	0.6148	0.7354	ns	yes
5b	AP5	Object A	0.6628	0.7179	ns	yes
		Object A*	0.6628	0.7179	ns	yes
	PBS	Object A	1.201	0.5486	ns	yes
		Object A*	1.201	0.5486	ns	yes
5c	AP5	Object A	0.2501	0.8825	ns	yes
		Object A*	0.2501	0.8825	ns	yes
	PBS	Object A	0.9246	0.6298	ns	yes
		Object A*	0.9246	0.6298	ns	yes
6e	ArchT	Object A	10.91	0.0043	**	no
		Object A*	10.91	0.0043	**	no
	eYFP	Object A	2.129	0.3448	ns	yes
		Object A*	2.129	0.3448	ns	yes
6f	ArchT	Object A	2.036	0.3612	ns	yes
		Object A*	2.036	0.3612	ns	yes
	eYFP	Object A	1.327	0.515	ns	yes
		Object A*	1.327	0.515	ns	yes
6g	ArchT	Object A	1.277	0.5282	ns	yes
		Object A*	1.277	0.5282	ns	yes
	eYFP	Object A	0.06228	0.9693	ns	yes
		Object A*	0.06228	0.9693	ns	yes
6h	ArchT	0.2331	0.89	ns	yes	
	eYFP	7.768	0.0206	*	no	
Supplementary 2b	Object A	0.3347	0.8459	ns	yes	
	Object A*	0.3347	0.8459	ns	yes	
Supplementary 2c	Object A	1.918	0.3833	ns	yes	
	Object A*	1.918	0.3833	ns	yes	
Supplementary 2e	NOL event	0.4908	0.7824	ns	yes	
	Circle	0.8324	0.6596	ns	yes	
Supplementary 4b	AP5	Object A	0.7775	0.6779	ns	yes
		Object A*	0.7775	0.6779	ns	yes
	PBS	Object A	1.94	0.379	ns	yes

		Object A*	1.94	0.379	ns	yes
Supplementary 4C	AP5	Object A	0.8876	0.6416	ns	yes
		Object A*	0.8876	0.6416	ns	yes
	PBS	Object A	1.929	0.3812	ns	yes
		Object A*	1.929	0.3812	ns	yes
Supplementary 4d	AP5	Object A	1.564	0.4576	ns	yes
		Object A*	1.564	0.4576	ns	yes
	PBS	Object A	0.1846	0.9118	ns	yes
		Object A*	0.1846	0.9118	ns	yes
Supplementary 4e	AP5	Object A	2.67	0.2631	ns	yes
		Object A*	2.67	0.2631	ns	yes
	PBS	Object A	1.199	0.549	ns	yes
		Object A*	1.199	0.549	ns	yes
Supplementary 7b	Object A		0.5594	0.756	ns	yes
	Object A*		0.5594	0.756	ns	yes
Supplementary 7c	Object A		0.4098	0.8147	ns	yes
	Object A*		0.4098	0.8147	ns	yes
Supplementary 7d	Object A		3.731	0.1548	ns	yes
	Object A*		3.731	0.1548	ns	yes

2. In general, it would be nice to have a little bit more detail on the methods and data analysis in the results section.

→ We believe we have explained the methods related to each part in the results section sufficiently for the reader to understand without being too redundant and we have provided the full details in the methods section for a more detailed and thorough explanation.

Explanation for the NOL behavior:

Line 73-79 Mice freely explored the location of two objects during a training session, and then during the test session one object was moved to a new location while the other remained in its original location (Supplementary Fig. 1). When mice recall the memory of object location, their exploration of the moved object is greater than that of the unmoved object. However, an equal exploration of the two objects indicates that the mice have not discriminated between the new and old locations and have been unable to retrieve the memory.

Explanation for the optogenetic experiment:

Line 101-107 A previously established and validated system in labeling specific events in CA3 neurons was used⁵, in which c-fos::fTA/KAI1::Cre double transgenic mice were injected with AAV₉-TRE-DIO-ChR2-mCherry into their hippocampal CA3 to specifically label the activated CA3 cells involved in learning the weak event with the blue-light sensitive ChR2. One day after training, mice were subjected to the test session, during which optical stimulation to their CA3-CA1 projections was conducted by shining blue light (473 nm) above the CA1 area at 20 Hz stimulation (Fig. 2a-c).

Explanation for the 2-training protocol:

Line 136-139 Mice underwent the weak NOL training, and then, 1 day later, repeated the same training again and were then tested for consolidation after one more day (Fig. 3a-c). Mice subjected to this two-training paradigm succeeded to recall the STM 1 day after re-exposure, which indicates consolidation of the STM trace (Fig. 3d).

Explanation for the pharmacological experiments:

Line 166-168 We repeated the two-training paradigm with a 1-day interval and injected anisomycin, a protein synthesis inhibitor, to the hippocampal CA1 region immediately after the first training session.

Line 193-195 Similar to the anisomycin experiment, using the two-training paradigm with a 1-day interval, mice injected with AP5, in the hippocampal CA1 region after either the first or second weak trainings

Explanation for the sleep deprivation experiment:

Line 215-217 To confirm the necessity of post-learning sleep in preserving the silent STM trace, we used the two-training paradigm with a 1-day interval in between, and subjected mice to a 5 h sleep deprivation immediately after the first training (Supplementary Fig. 7a-c).

Explanation for the optogenetic inhibition experiment:

Line 221-226 a group of mice was injected with AAV₉-CaMKII-ArchT-eYFP (test group) and another group was injected with AAV₁-CaMKII-eYFP (control group). Orange light (589 nm) was delivered to their hippocampal CA1 region during all the NREM sleep stages that appeared within a 2-hour sleep session immediately after the first training to optogenetically silence the offline hippocampal CA1 neurons.

3. How many mice were used for the calcium imaging experiments?

→ As previously explained in point 1, we have omitted the calcium imaging experiment from our revised manuscript, however, a total of 5 mice was used.

4. Can you give more details on what is the logic of injecting an AAV into CA3 and silencing CA1 (in Figure 2), other than just mentioning that this system was previously validated. In general, I would like to get more details on why CA1 was chosen to maintain STM as compared to other potential areas.

→ The CA1 was chosen for our investigation because the NOL task is a spatial memory task in which the hippocampus is directly involved⁷.

→ This approach was used in order to define a clear neuronal pathway at which the STM silent engram is stored. Accordingly, we had a choice between targeting the Entorhinal cortex-CA1 pathway (Perforant pathway) or the CA3-CA1 pathway (Schaffer collaterals pathway).

→ We had already established and validated an engram-labelling system for the CA3 neurons in our lab and so we used this system in labelling the CA3 neurons and activating their terminals in the CA1 to specifically reveal whether this pathway stores the STM silent engram or not.

→ Had this pathway given a negative result, we would have moved to the other pathway (Entorhinal cortex-CA1), however, since the CA3-CA1 pathway appeared to be involved in this storage, we did not feel the need to explore the involvement of other neuronal pathways within the hippocampus in the scope of our study.

And discuss other areas that may participate in the engram of STM (linked to major point 2)

→ As mentioned in response to major point 2, we have edited our discussion section to better relate to the other STM studies as mentioned by the reviewer.

Line 303-307 Given the diversity of STM-inducing behavioral paradigms other than the NOL and the non-hippocampal forms of STM stored in many cortical regions, especially those that are part of the sensorimotor processing^{47,48} or those that are part of planning⁴⁹, future studies are needed to elucidate whether other types of STM also have a long-term silent trace regardless of their origin or function in the brain.

5. The model in Supp figure 8 is not very clear for me. The graph requires more description. Also, once better explained, I think this would be very informative in the main results.

→ We have edited Supplementary Fig. 8 to make it clearer.

→ Briefly, we have omitted the terms “memory reactivation” after removing the calcium imaging data.

→ We have added the sign (+) to indicate that these are the requirements for the storage of the silent STM and for its subsequent activation.

Modified Supplementary Fig. 8

→ We have also edited the figure legend for better explanation.

Supplementary Figure 8. Hypothesized model for STM vs. LTM engrams. One strong event is sufficient to consolidate an LTM, which is stored in the form of an active engram that can be naturally recalled. One weak event is not sufficient for consolidation; however, it forms an STM stored in the form of a silent engram and this storage requires post-learning NMDAR activation, new protein synthesis, and NREM sleep. A repeated weak event can activate this silent engram by consolidation within its lifetime (<6 days) and this activation process requires post-learning NMDAR activation as well. However, if the second weak event is repeated after 6 days, the first STM engram is no longer available and the repeated weak event is then processed as a new event by forming a new silent STM engram with similar properties. This model does not take into account the initial period of transient same-day STM recall, but rather focuses on the detailed long-term storage of STM for the sake of clarity.

→ We hope that now the modified figure and its modified legend combined with our explanation in the discussion section are understandable to the reviewer and the readers.

References:

- 1 Josselyn, S. A. & Tonegawa, S. Memory engrams: Recalling the past and imagining the future. *Science* **367** (2020).
- 2 Kitamura, T. *et al.* Engrams and circuits crucial for systems consolidation of a memory. *Science* **356**, 73-78 (2017).
- 3 Tonegawa, S., Morrissey, M. D. & Kitamura, T. The role of engram cells in the systems consolidation of memory. *Nature Reviews Neuroscience* **19**, 485-498 (2018).
- 4 Gilad, A., Gallero-Salas, Y., Groos, D. & Helmchen, F. Behavioral strategy determines frontal or posterior location of short-term memory in neocortex. *Neuron* **99**, 814-828. e817 (2018).
- 5 Goard, M. J., Pho, G. N., Woodson, J. & Sur, M. Distinct roles of visual, parietal, and frontal motor cortices in memory-guided sensorimotor decisions. *elife* **5** (2016).
- 6 Li, N., Chen, T.-W., Guo, Z. V., Gerfen, C. R. & Svoboda, K. A motor cortex circuit for motor planning and movement. *Nature* **519**, 51-56 (2015).
- 7 Winters, B. D., Forwood, S. E., Cowell, R. A., Saksida, L. M. & Bussey, T. J. Double dissociation between the effects of peri-postrhinal cortex and hippocampal lesions on tests of object recognition and spatial memory: heterogeneity of function within the temporal lobe. *Journal of Neuroscience* **24**, 5901-5908 (2004).

Reviewers' comments:

Reviewer #1 (Remarks to the Author):

The manuscript has improved significantly, primarily on account of the information specificity control in the supplementary material. This is robust and welcome.

However, the removal of the calcium imaging analysis is a significant loss to the manuscript's body.

I am still unconvinced as to the sleep experiments' place in this manuscript. The NMDA and anisomycin experiments are good and make sense, but the sleep experiments seem quite orthogonal.

At a conceptual level, the term "silent engram" is not well justified. Just because some authors have used this term does not mean it is well established. Even if it is, this is not what the authors have in this manuscript. Tonegawa and colleagues have used "silent engram" to refer to an amnesic engram that cannot be accessed by natural cues. In the case of this manuscript, clearly the engram can be accessed by natural cues. So, it is not a silent engram it just an engram that does not warrant behavioral changes in the conditions given to the subjects. This is not a major problem with the manuscript, but I believe it essential to correct this prior to any publication. This also affects the model the authors present.

Otherwise, no major issues with the revised manuscript.

Reviewer #2 (Remarks to the Author):

Although the authors have decided to omit figure 6 from the current version of the manuscript, I think there is plenty of interesting and valuable results even without the calcium imaging data. Other than that, all my concerns were answered and I fully support publication. Congratulations on a great study.

Response to reviewers: 2nd revision

We thank the reviewers for taking more time to thoroughly revise our manuscript and for providing us with valuable and constructive comments. We have addressed all of the reviewers' comments and provide our point- by-point response to the comments below. We have revised our manuscript, guided by the reviewers' feedback, **and all corrections or additions made to the text are colored in red**. We hope that our revised manuscript is now suitable for publication in *Communications Biology*.

Reviewer #1 (Remarks to the Author):

The manuscript has improved significantly, primarily on account of the information specificity control in the supplementary material. This is robust and welcome. However, the removal of the calcium imaging analysis is a significant loss to the manuscript's body.

→ We thank the reviewer for their valuable comments and revision which helped us reshape our manuscript in its final form and present it to the scientific community the best way we can.

1) I am still unconvinced as to the sleep experiments' place in this manuscript. The NMDA and anisomycin experiments are good and make sense, but the sleep experiments seem quite orthogonal.

→ We thank the reviewer for their opinion, we explain below the justification for our sleep experiment then we explain the modifications done to the manuscript to better clarify this justification.

→ Justification for sleep experiments in our manuscript:

- Our manuscript focuses on the potential existence of a short-term memory trace and its long-term storage.
- Our investigation of the possible requirements for this storage is guided by the well-known prerequisites of storage for long-term memory traces, a process known as memory consolidation, which as we highlight in our manuscript are:
 - New protein synthesis ¹⁻⁴
 - Synaptic plasticity through NMDAR activation ⁴⁻⁶
 - A period of post-learning sleep ⁷⁻¹² including both NREM ¹³ and REM ¹⁴ stages.
- Guided by these well-acknowledged facts by the scientific community, we carried out our mechanistic investigations on the long-term storage of the short-term memory trace mainly in 3 lines of experiments focusing on the 3 main requirements for consolidation:
 - a) Whether protein synthesis is required for this storage → Anisomycin experiment (Fig. 4)
 - b) Whether NMDAR activation is required for this storage → AP5 experiments (Fig. 5)
 - c) Whether a period of post-learning sleep is required for this storage → sleep experiments (Fig. 6).
- Furthermore, we used not one but two approaches to demonstrate the necessity of sleep for the long-term storage of the short-term memory trace:

- a) We performed a sleep deprivation experiment to demonstrate a general requirement for post-learning sleep, regardless of a specific sleep stage.
 - b) We performed a specific optogenetic silencing experiment to demonstrate a causal requirement for NREM sleep in specific.
 - The results obtained from the two sleep experiments complement each other and provide evidence into a crucial requirement for sleep in the long-term storage of a short-term memory similar to its requirement for the storage of a long-term memory.
- Furthermore, as we now refer to in our discussion:
 - A cross-link exists between sleep and protein synthesis where a period of post-learning sleep has been shown to be essential for protein synthesis ¹⁵⁻¹⁹.
 - Another bi-directional cross-link exists between sleep and NMDAR activation where a period of post-learning sleep has been shown to be essential for synaptic plasticity ^{20,21} and altering NMDAR activation in return affects sleep ^{22,23}.
 - Taken together, guided by these results from literature, it does not seem odd to investigate in our study the requirement for a period of post-learning sleep in the long-term storage of the short-term memory trace after we have shown the requirement for both protein synthesis and NMDAR activation.
 - Finally, we believe the results obtained from the three lines of experiments (Anisomycin, AP5 and sleep) provide an integrated picture and unprecedented insight into the requirements of those three pillars in the long-term storage of short-term memory traces.

References:

- 1 Davis, H. P. & Squire, L. R. Protein synthesis and memory: a review. *Psychological bulletin* **96**, 518 (1984).
- 2 Abel, T. *et al.* Genetic demonstration of a role for PKA in the late phase of LTP and in hippocampus-based long-term memory. *Cell* **88**, 615-626 (1997).
- 3 Kandel, E. R., Dudai, Y. & Mayford, M. R. The molecular and systems biology of memory. *Cell* **157**, 163-186 (2014).
- 4 Korte, M. & Schmitz, D. Cellular and system biology of memory: timing, molecules, and beyond. *Physiological reviews* **96**, 647-693 (2016).
- 5 Lüscher, C. & Malenka, R. C. NMDA receptor-dependent long-term potentiation and long-term depression (LTP/LTD). *Cold Spring Harbor perspectives in biology* **4**, a005710 (2012).
- 6 Paoletti, P., Bellone, C. & Zhou, Q. NMDA receptor subunit diversity: impact on receptor properties, synaptic plasticity and disease. *Nature Reviews Neuroscience* **14**, 383-400 (2013).
- 7 Diekelmann, S. & Born, J. The memory function of sleep. *Nature Reviews Neuroscience* **11**, 114-126 (2010).
- 8 Rasch, B. & Born, J. About sleep's role in memory. *Physiological reviews* (2013).
- 9 Areal, C. C., Warby, S. C. & Mongrain, V. Sleep loss and structural plasticity. *Current opinion in neurobiology* **44**, 1-7 (2017).
- 10 Havekes, R. & Abel, T. The tired hippocampus: the molecular impact of sleep deprivation on hippocampal function. *Current opinion in neurobiology* **44**, 13-19 (2017).

- 11 Raven, F., Meerlo, P., Van der Zee, E. A., Abel, T. & Havekes, R. A brief period of sleep deprivation causes spine loss in the dentate gyrus of mice. *Neurobiology of learning and memory* **160**, 83-90 (2019).
- 12 Heckman, P. R., Kuhn, F. R., Meerlo, P. & Havekes, R. A brief period of sleep deprivation negatively impacts the acquisition, consolidation, and retrieval of object-location memories. *Neurobiology of learning and memory* **175**, 107326 (2020).
- 13 Girardeau, G., Benchenane, K., Wiener, S. I., Buzsáki, G. & Zugaro, M. B. Selective suppression of hippocampal ripples impairs spatial memory. *Nature neuroscience* **12**, 1222-1223 (2009).
- 14 Boyce, R., Glasgow, S. D., Williams, S. & Adamantidis, A. Causal evidence for the role of REM sleep theta rhythm in contextual memory consolidation. *Science* **352**, 812-816 (2016).
- 15 Seibt, J. *et al.* Protein synthesis during sleep consolidates cortical plasticity in vivo. *Current Biology* **22**, 676-682 (2012).
- 16 Tudor, J. C. *et al.* Sleep deprivation impairs memory by attenuating mTORC1-dependent protein synthesis. *Science signaling* **9**, ra41-ra41 (2016).
- 17 Lyons, L. C., Chatterjee, S., Vanrobaeys, Y., Gaine, M. E. & Abel, T. Translational changes induced by acute sleep deprivation uncovered by TRAP-Seq. *Molecular brain* **13**, 1-18 (2020).
- 18 Gaine, M. E. *et al.* Altered hippocampal transcriptome dynamics following sleep deprivation. *Molecular brain* **14**, 1-17 (2021).
- 19 Raven, F. *et al.* Elucidating the role of protein synthesis in hippocampus-dependent memory consolidation across the day and night. *European Journal of Neuroscience* **54**, 6972-6981 (2021).
- 20 Kopp, C., Longordo, F., Nicholson, J. R. & Lüthi, A. Insufficient sleep reversibly alters bidirectional synaptic plasticity and NMDA receptor function. *Journal of Neuroscience* **26**, 12456-12465 (2006).
- 21 McDermott, C. M., Hardy, M. N., Bazan, N. G. & Magee, J. C. Sleep deprivation-induced alterations in excitatory synaptic transmission in the CA1 region of the rat hippocampus. *The Journal of physiology* **570**, 553-565 (2006).
- 22 Chen, C., Hardy, M., Zhang, J., LaHoste, G. J. & Bazan, N. G. Altered NMDA receptor trafficking contributes to sleep deprivation-induced hippocampal synaptic and cognitive impairments. *Biochemical and biophysical research communications* **340**, 435-440 (2006).
- 23 Burgdorf, J. S. *et al.* NMDAR activation regulates the daily rhythms of sleep and mood. *Sleep* **42**, zsz135 (2019).

→ Modifications done to the revised manuscript to better clarify this justification:

We include the above mentioned justification in the manuscript at several locations with the addition of new references to better clarify to the audience how the sleep experiments are connected to the rest of the manuscript and the other performed experiments.

- In the introduction:

Lines 49 to 58

Memory consolidation process depends on **several factors such as** new protein synthesis, whereby the inhibition of post-learning protein synthesis blocks LTM but not STM retrieval^{25,26}. **It also depends on N-methyl-D-aspartate receptor (NMDAR) activation, where synaptic plasticity has been shown to be integral for memory consolidation^{5,27,28}. Finally,** consolidation has also been shown to occur after a period of post-learning sleep²⁹⁻³⁴ which consists of two main stages, non-rapid eye movement sleep (NREM) characterized by slow delta waves, spindles and sharp wave-ripples (SWRs), and rapid-eye sleep (REM) characterized by fast theta rhythms³⁰. It has been shown that hippocampal SWRs, which occur during NREM sleep, and theta rhythms, which occur during REM sleep, are both critical for the consolidation of spatial memories^{35,36}.

- In the results section:

Lines 163 to 165

Similar to protein synthesis and NMDAR activity; post-learning sleep has also been shown to be integral for the consolidation of long-term memories²⁹⁻³⁴. Accordingly, we tested the potential requirement for sleep in the long-term storage of the aforementioned STM trace.

-In the discussion section:

Lines 222 to 223

Furthermore, **a cross link has been shown to exist between sleep and protein synthesis where** sleep deprivation has been shown to impair hippocampal protein synthesis and consequently memory consolidation⁴⁸⁻⁵².

Lines 227 to 229

Another cross link exists between sleep and NMDAR activity where a period of post-learning sleep has been shown to be essential for synaptic plasticity^{53,54} and altering NMDAR activation in return affects sleep^{55,56}.

2) At a conceptual level, the term "silent engram" is not well justified. Just because some authors have used this term does not mean it is well established. Even if it is, this is not what the authors have in this manuscript. Tonegawa and colleagues have used "silent engram" to refer to an amnesic engram that cannot be accessed by natural cues. In the case of this manuscript, clearly the engram can be accessed by natural cues. So, it is not a silent engram it just an engram that does not warrant behavioral changes in the conditions given to the subjects. This is not a major problem with the manuscript, but I believe it essential to correct this prior to any publication. This also effects the model the authors present. Otherwise, no major issues with the revised manuscript.

→ We thank the reviewer for their criticism; we have complied with this request and have removed the term "silent engram" from our interpretations.

→ Accordingly, the manuscript has been modified where we now refer to our found engram simply as "STM engram" instead of "silent STM engram".

→ We have modified the title of our manuscript as a result by removing the word "silent".

→ We have also modified our model figure as follows:

Reviewer #2 (Remarks to the Author):

Although the authors have decided to omit figure 6 from the current version of the manuscript, I think there is plenty of interesting and valuable results even without the calcium imaging data. Other than that, all my concerns were answered and I fully support publication. Congratulations on a great study.

→ We thank the reviewer for their valuable comments and revision which helped us reshape our manuscript in its final form and present it to the scientific community the best way we can.